# Can Urbanization-Driven Land-Use and Land-Cover Change Reduce Ecosystem Services? A Case of Coupling Coordination Relationship for Contiguous Poverty Areas in China

**Jian Zhang** [1,†] ⓘ, **Xin Lu** [1,†], **Yao Qin** [1], **Yuxuan Zhang** [1] **and Dewei Yang** [2,*] ⓘ

1   College of Geography and Environmental Science, Northwest Normal University, Lanzhou 730070, China; jianzhang@nwnu.edu.cn (J.Z.); 2021212899@nwnu.edu.cn (X.L.); 2021212884@nwnu.edu.cn (Y.Q.); 2021212893@nwnu.edu.cn (Y.Z.)
2   School of Geographical Sciences, Southwest University, Chongqing 400715, China
*   Correspondence: younglansing@gmail.com
†   These authors contributed equally to this work.

**Abstract:** New urbanization often leads to land-use and land-cover change (LUCC), which inevitably affects ecosystem services (ESs). Although it is traditionally believed that urbanization reduces ecosystem services, some studies have shown that reasonable urban development facilitates ecosystem conservation. Previous studies have focused on the impacts of urbanization on either LUCC or ESs, with fewer dynamic assessments of the coordination of the three. Taking China's contiguous poor areas (CPAs) as an example, this study applied coupling coordination, path analysis, and a multiscale geographically weighted regression (MGWR) model to identify the dynamic relationship among urbanization, land use, and the environment and then predicted their coupling coordination under shared socioeconomic pathways (SSP-RCP) in 2035 using the Patch Generation Land Use Simulation (PLUS) and a random forest model. The results of the study show that (1) urbanization, land-use change, and environmental loads in China's CPAs showed an inconsistent upward trend. There was a slight overall decrease in ESs before 2013, which was consistent with the early stage of the Environmental Kuznets Curve (EKC); after that time, they showed different characteristics. (2) From 2000 to 2018, the coupling coordination degree of CPAs decreased slightly due to urbanization, geographic factors, and grassland and unused land. LUCC was essential to maintaining the system balance. The SN (southern contiguous poverty area) was at a basic level of coordination, while the other regions showed a moderate imbalance. (3) According to scenario projections, the degree of coupling coordination in all regions will increase by 2035. Environmental prioritization and sustainable routes are the best options for CPAs' development. The SN is more stable, while the WN (western contiguous poverty area) has the lowest coupling coordination. (4) Environmentally friendly urbanization should be carried out with land management tailored to local conditions. Measures that could be recommended include establishing ecological pilot zones in SN areas, prioritizing the protection of grassland ecosystems in WN areas, and promoting intensive land use in the NN (northern contiguous poverty area). The present study offers a novel perspective on the interplay between the economy and the environment at the county level and achieves predictive coupling coordination through the integration of PLUS and random forest models. This investigation of coordinated urbanization–LUCC–ES development in CPAs yields valuable insights for enhancing environmental and economic well-being in similar regions within China, as well as globally.

**Keywords:** urbanization; LUCC; ESs; SSP-RCP scenarios; CPAs

## 1. Introduction

Currently, urbanization and poverty alleviation are two critical aspects that are inextricably linked to the world's sustainable development. Despite the ongoing global

phenomenon of urbanization, poverty remains a multifaceted global problem [1,2]. Urbanization is a multifaceted process that can be studied from a variety of angles, including population, economic, spatial, and other considerations [3]. Essentially, urbanization causes population concentration in large towns and cities, as well as changes in land use and land cover (LUCC) and the economic structure. According to the World Urbanization Prospects, developing-country urbanization has more than doubled from 17.6% in 1950 to 56% in 2021. However, according to a United Nations survey, approximately 1.3 billion people are estimated to be living in poverty worldwide, with the majority of them concentrated in developing countries. The process of urbanization frequently coincides with a decrease in poverty rates, particularly in rural areas [4,5]. Meanwhile, urbanization creates challenges such as land scarcity and environmental degradation, both of which impede long-term development. Specifically, there is a specific inverted U-shaped relationship between urbanization and the poverty rate [6,7]. An urbanization level of around 40% is conducive to poverty alleviation in Vietnam, whereas excessive urbanization will deepen poverty [8]. The relationship between urbanization and poverty alleviation is complex and influenced by the stage of development and regional characteristics. Therefore, further research is necessary to investigate urbanization development in impoverished areas.

Urbanization is a major driver of LUCC [9,10]. The expansion of urban areas results in the appropriation of agricultural and ecological lands, resulting in environmental degradation and a decline in ecosystem services (ESs) [11,12]. Due to the unsustainable use of land resources, developing countries experiencing rapid urbanization face critical challenges in achieving sustainable development [13]. The process of urbanization has significantly altered ESs through land overexploitation [14]. According to a study conducted in the Solomon Islands, the process of urbanization exerts a detrimental influence on ESs by reducing cropland and forests [15]. Predictions from around the world find that urban sprawl and development will continue to wreak havoc on ESs [16,17]. Recent studies have confirmed the close relationship between urbanization, LUCC, and ESs and found the importance of land use [18]. Cropland, water resource availability, ecological land, and forest cover are important ecological factors for coupling coordination [19]. However, research has mostly focused on urbanization or LUCC in relation to ESs in terms of cities. There is a lack of studies on poorer areas and smaller administrative units and even fewer studies on the linkages between the three.

The negative correlation between urbanization, LUCC, and ESs, however, is not absolute. The EKC theory suggests that environmental degradation initially increases with economic growth. After reaching a critical point, or a "tipping point", environmental pollution declines as income continues to increase [20]. The formulation of this theory quickly attracted the attention of researchers and has gradually expanded to include urban development, human well-being, and environmental sustainability [21]. Latin America and the Caribbean, East Asia and the Pacific, and Europe and Central Asia, among others, exhibit an inverted "U"-shaped link between the ecological environment and economic growth [19,22]. However, the EKC hypothesis is not supported in Sub-Saharan Africa and some countries of the Belt and Road Initiative (BRI) in terms of carbon emissions [22,23]. The relationship between the economy and the environment remains highly uncertain [24–26]. Research shows that this hypothesis depends on the merit of natural resources [27]. Countries need to make appropriate policy choices to ensure environmental sustainability while they strive to develop their economies. The short-term effects of the presence of EKCs in member countries under China's BRI are not effective in the long term [23]. The question of achieving economic growth and urbanization without sacrificing the environment remains an important issue for developing countries. Policy regulations can partly mitigate the adverse effects of the early stages of development [27]. China declared that contiguous poverty areas (CPAs) would be the main battleground for poverty alleviation in 2011 and then launched poverty alleviation policies after releasing a list of targeted counties and cities from 2012 to 2013 [28,29]. Finally, the complete abolition of poverty was achieved in 2020. Furthermore, China issued the "New Urbanization Plan

(2014–2020)" in 2014 and proposed the "Memorandum of Opinions on Establishing and Improving the Institutional Mechanism and Policy System of Urban-Rural Integration Development" in 2019, with the goal of coordinating urban–rural integration and resource allocation [30]. Despite some positive developments, rapid urbanization continues to pose ecological security challenges to CPAs [31,32], including environmental degradation [33,34], land overexploitation [35,36], etc. For example, Wang et al. [37] discovered that changes in LUCC caused by urbanization would significantly worsen ESs across townships in Central Yunnan. Therefore, it is imperative to urgently address the ecological threats posed by urbanization in CPAs to achieve coordinated development. However, there are fewer case studies on CPAs. Most of the studies focus on smaller regions.

The proposed creation of a resource-efficient and environmentally friendly urbanization development model introduces new challenges to the coordinated development of urbanization–LUCC–ESs [30]. Some scholars have already been concerned with the subject of spatial and temporal changes in urbanization, LUCC, and ESs. However, most previous research has concentrated on country- and city-level studies, and the understanding of the complex relationship between urbanization, LUCC, and ESs remains inadequate [38–42]. Moreover, there is a notable absence of coordinated monitoring and prediction in poverty areas. Therefore, this paper aims to scrutinize CPAs to explore the environmental impacts, as portrayed by ESs, within the urbanization development process. It seeks to integrate the PLUS model with a random forest model to forecast the coupling coordination degree of urbanization–LUCC–ESs. The study aims to (1) identify the spatiotemporal variation in urbanization, LUCC, and ESs within CPAs; (2) assess the influencing factor of the coupling coordination degree among urbanization, LUCC, and ESs using path analysis and multi-scale geographically weighted regression (MGWR); (3) predict the coupling coordination degree in alternative shared socioeconomic pathway/representative concentration pathway (SSP-RCP) scenarios by 2035. This study will provide new perspectives for the coordinated development of regional urbanization, LULC, and ESs in economically disadvantaged areas. Moreover, it offers a scientifically grounded approach and novel methodology for informing decision-making processes related to regional sustainable development.

## 2. Materials and Methods

### 2.1. Study Area

The CPAs cover over 30% of China's land area and are made up of 14 regions and 680 counties. Longitudes 73.50° and 126.98° E and latitudes 21.14° and 48.53° N define the region. The altitude ranges from 6 to 8689 m, with a clear westward trend and an eastward trend. CPAs' complex terrain contributes to a wide range of climates. The southern region is hot and humid, whereas the northern region is cooler and has lower precipitation rates. CPAs had a population of 3.13 million people in 2019, with a per capita disposable income of about USD 1634, which is less than 20% of the national average of about USD 10,103 [43]. As a result, CPAs are critical areas of focus in China's efforts to alleviate poverty. Figure 1 shows the general location of CPAs.

### 2.2. Research Framework

This study calculated urbanization, LUCC, and ES indicators based on remote sensing and statistical data and calculated the coupling coordination degree among the three [16,44,45]. The spatial and temporal relationships between urbanization, LUCC, and ESs in CPAs were explored using MGWR and path analysis methods [46–48]. Finally, the PLUS [49,50] and random forest [51] models were combined to assess the future coupling coordination of CPAs under SSP-RCP scenarios. The research framework is illustrated in Figure 2.

### 2.3. Data Collection

This study included, among other variables, land-use data, climate data, and economic data. All data were resampled to a resolution of 1 km × 1 km. Table 1 contains a comprehensive list of data sources and additional information.

*2.4. Methods*

2.4.1. LUCC Transfer Matrix

The LUCC transfer matrix reflects changes in land area for various types over a specific time period within a specific region. The research looked at LUCC in 2000, 2005, 2013, 2015, and 2018 (the change rate of more than two land-use types in the same year was more than three times the benchmark *K*-value). The LUCC transfer matrix is typically represented [52] in the following format:

$$X = \begin{bmatrix} X_{11} & X_{12} & \cdots & X_{1j} \\ X_{21} & X_{21} & \cdots & X_{2j} \\ \vdots & \vdots & \ddots & \vdots \\ X_{i1} & X_{i2} & \cdots & X_{ij} \end{bmatrix} \tag{1}$$

$$K = \frac{U_b - U_a}{U_a} \times \frac{1}{T} \times 100\% \tag{2}$$

$X_{ij}$ represents the land area of type *i* that was converted to type *j*; *K* is the annual rate of change during the period; $U_a$ and $U_b$ represent the total area of land use; and *T* is the length of time measured in years.

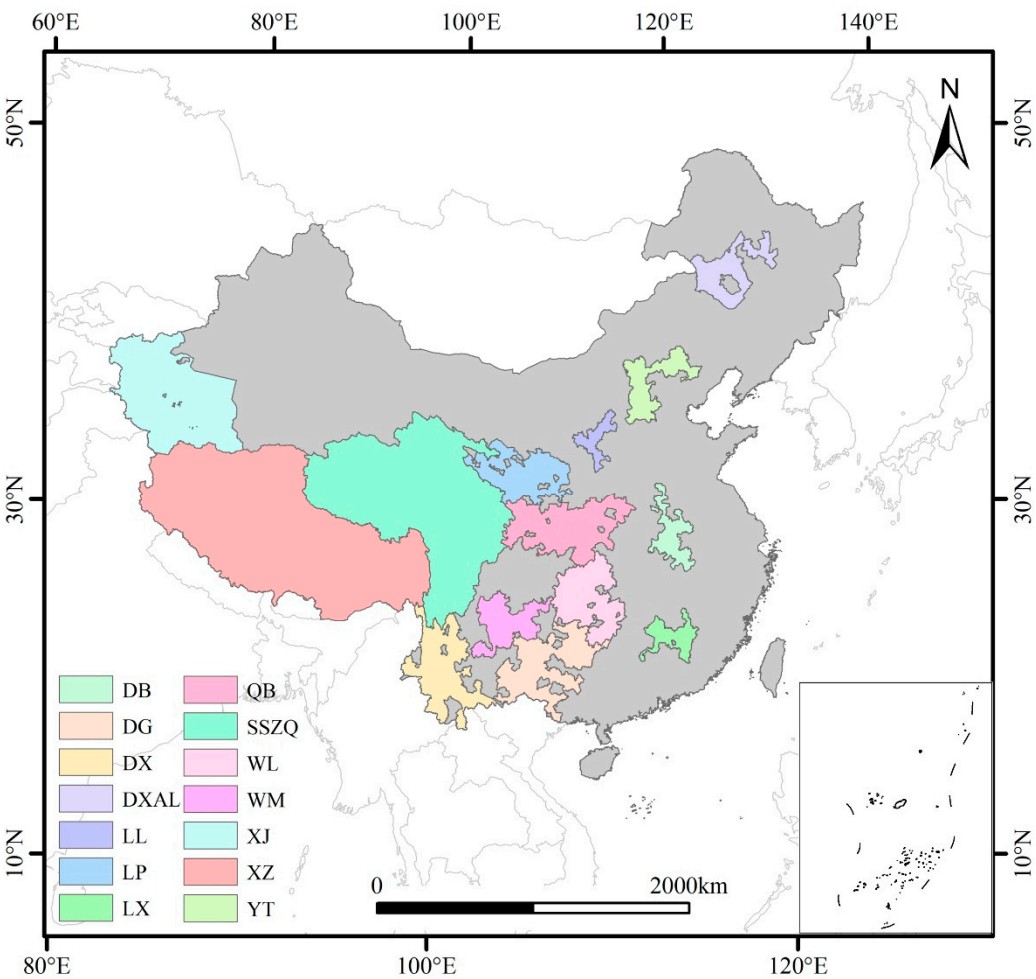

**Figure 1.** The locations of CPAs. LP is Liupan Mountains; QB is Qinba Mountains; WL is Wuling Mountains; WM is Wumeng Mountains; DG is Yunnan–Guizhou–Guizhou Rocky Desertification Areas; DX is Western Yunnan Border Mountains; DXAL is Southern Foot Mountains of the Great Xing'an Mountains; YT is Yanshan-Taihang Mountains; LL is Luliang Mountains; DB is Dabie Mountains; LX is Luoxiao Mountains; XZ is Tibet; SSZQ is the Tibetan Region in the four provinces, and XJ is the four prefectures in southern Xinjiang Region.

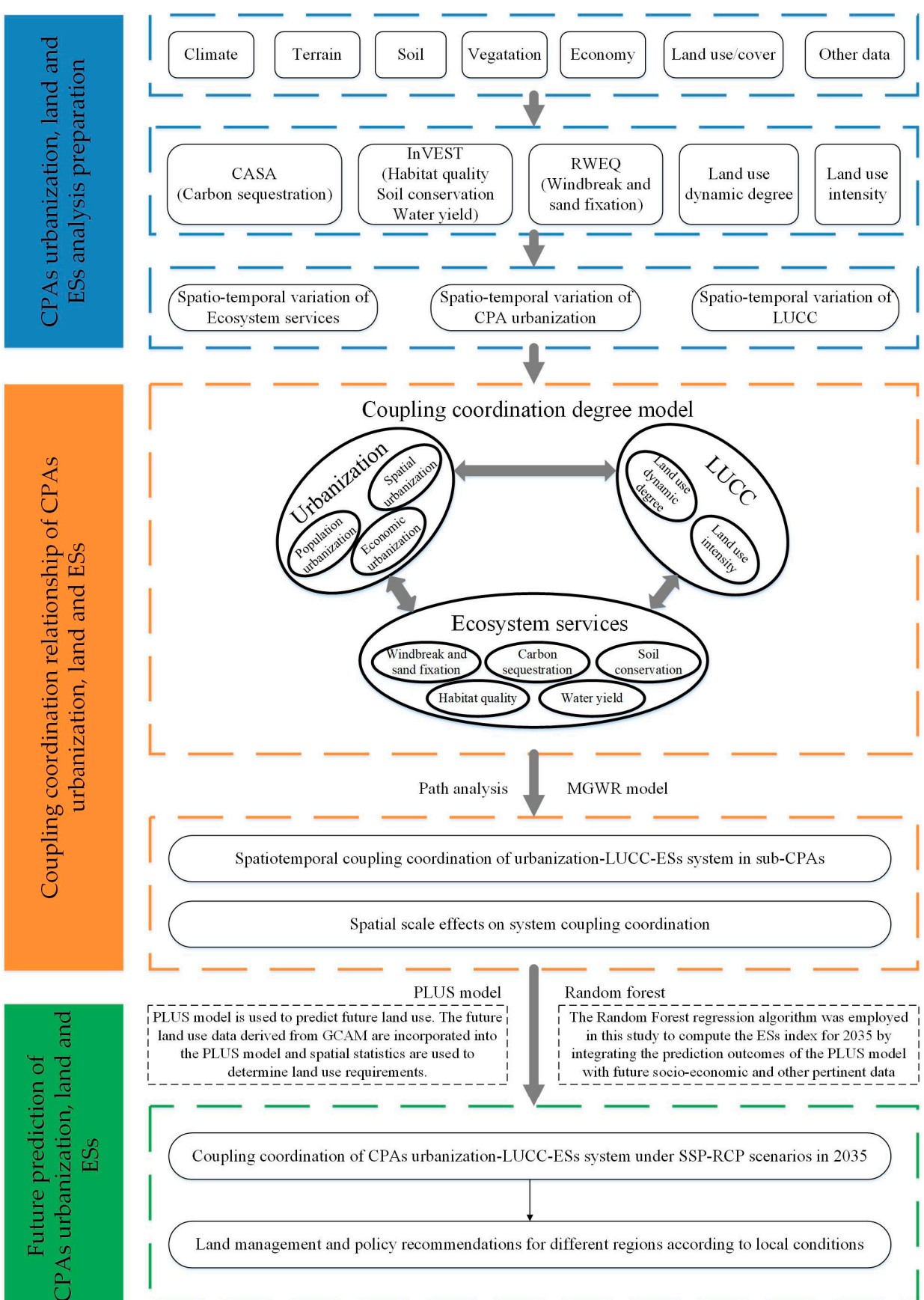

**Figure 2.** Research framework of coupling coordination for urbanization–LUCC–ESs in CPAs.

**Table 1.** Data sources.

| Category | Data | Year | Resolution | Data Resource |
|---|---|---|---|---|
| Land-use and land-cover data Future land requirements data | Land-use data | 2000–2020 2035 | 30 m 1 km | CLCD (https://zenodo.org/ (accessed on 11 April 2022)) GCAM-Demeter land use dataset at 0.05-degree resolution (https://data.pnnl.gov/group/nodes/dataset (accessed on 7 December 2022)) |
| Terrain | DEM | 2000 | 250 m | China Resources and Environmental Science and Data Center (https://www.resdc.cn/ (accessed on 13 October 2022)) |
| Climate (monthly) | Temperature Precipitation Potential evapotranspiration Surface solar radiation | 2000, 2005, 2013, 2015, 2018 | 50 km 10 km | CRU (https://crudata.uea.ac.uk/cru/data/ (accessed on 13 October 2022)) National Qinghai-Tibet Plateau Scientific Data Center (http://data.tpdc.ac.cn/ (accessed on 21 September 2022)) |
| Climate (daily) | Temperature Precipitation Snow depth Wind speed | 2000, 2005, 2013, 2015, 2018 | 1 km | National Centers for Environmental Information (https://www.ncei.noaa.gov/ (accessed on 25 September 2022)) National Qinghai-Tibet Plateau Scientific Data Center (http://data.tpdc.ac.cn/ (accessed on 25 September 2022)) |
| Soil | Soil data | 2008 | 1 km | HWSD v1.2 (http://www.fao.org/ (accessed on 25 September 2022)) |
| Vegetation | NDVI | 2000, 2005, 2013, 2015, 2018 | 1 km | Resource and Environmental Science and Data Center (http://www.resdc.cn/ (accessed on 2 October 2022)) |
| Economy (future) | Population GDP Added value of primary, secondary, and tertiary industries | 2035 | 1 km | Gridded datasets for population and economy under Shared Socioeconomic Pathways for 2020–2100 (https://china.scidb.cn (accessed on 18 June 2023)) |
| Economy (history) | Population Nighttime light data GDP Added value of primary, secondary, and tertiary industries | 2000, 2005, 2013, 2015, 2018 | 1 km 0.05 degree 1 km | Worldpop (https://www.worldpop.org/ (accessed on 26 November 2022)) Global NPP-VIIRS-like nighttime light data Version 3.1 (https://dataverse.harvard.edu/ (accessed on 26 November 2022)) Gridded global datasets for Gross Domestic Product and Human Development Index (https://datadryad.org/stash/dataset (accessed on 26 November 2022)) China County Statistical Yearbook (https://www.stats.gov.cn/ (accessed on 27 June 2023)) |
| The road data | Highway network data Railway network data | 2021 | | OpenStreetMap (https://www.openstreetmap.org/ (accessed on 26 November 2022)) |
| Validation data | ESs | 2010 | 250 m | Spatial Dataset of Ecosystem Services in China 2010 (https://www.scidb.cn/ (accessed on 14 May 2023)) |

Notes: Land-use and land-cover data were adjusted to better align shrubs with forests, snow with unused land, and wetlands with water for future land-use models.

### 2.4.2. Index System Construction

This study constructed an index system using several secondary indicators of urbanization, LUCC, and ESs. Table 2 lists the specific indicators and weights. The standardization and weighting methods are shown in Equations (14)–(17). The verification results of ESs are presented in Figure S1.

The data used for this study range in size and magnitude, and they include both positive and negative factors. The normalization method is calculated as follows:

$$X_i = \frac{x_i - x_{min}}{x_{max} - x_{min}} \tag{14}$$

$$X_j = \frac{x_{max} - x_j}{x_{max} - x_{min}} \tag{15}$$

where $X_{i(j)}$ is the normalization result; $x_i$ is the positive indicator; and $x_j$ is the negative indicator.

**Table 2.** Urbanization index system.

| System | Index | Variables and Formulas | | Weights | |
|---|---|---|---|---|---|
| Urbanization system | Population urbanization (0.33) | Percentage of population in urban areas | | 0.46 | |
| | | Population density | | 0.54 | |
| | Economic urbanization (0.33) | Per capita GDP | | 0.43 | |
| | | The percentage of the added value of secondary industry | | 0.35 | |
| | | The percentage of the added value of tertiary industry | | 0.21 | |
| | Spatial urbanization (0.33) | Percentage of construction land | | 1 | |
| LUCC system | Land-use dynamic degree | $LD = \frac{\sum_{i=1}^{n} VLU_{I-J}}{2\sum_{i=1}^{n} LU_i} \times \frac{1}{T} \times 100\%$ | (3) | 0.69 | [53] |
| | | $LD$ is land-use dynamic degree; $LU_i$ is the area of type $i$ at the starting time; $LU_{i-j}$ is the area of land-use types from $i$ to $j$ during the period, and $T$ is study time. $T$ is set in years. | | | |
| | Land-use intensity | $LI = \sum\limits_{i=1}^{n} \frac{A_i}{A} \times C_i$ | (4) | 0.31 | [54] |
| | | $LI$ is the comprehensive land-use intensity; $A_i$ is the area of type $i$ at the starting time; $A$ is the area of land-use types from $i$ to $j$; $C_i$ is the land-use intensity assignment for site type $i$. | | | |
| ESs system | SC (InVEST) | $SR = RKLS - USLE$ | (5) | 0.49 | |
| | | $USLE = R \times K \times LS \times C \times P$ | (6) | | |
| | | $RKLS = R \times R \times K \times LS$ | (7) | | |
| | | $SR$ is the SC amount (t·hm$^{-2}$); USLE is sediment retention (t·hm$^{-2}$); RKLS is the potential soil erosion (t·hm$^{-2}$); $R$ is the rainfall erosivity (MJ·mm·hm$^{-2}$·h$^{-1}$); $K$ is the soil erodibility factor (t·hm$^2$·h·MJ$^{-1}$·mm$^{-1}$); $LS$, $C$, and $P$ represent the slope length gradient, vegetation coverage, and erosion management, respectively (dimensionless). | | | [55] |
| | WY (InVEST) | $Y_x = (1 - AET_x/P_x) \times P_x$ | (8) | 0.1 | |
| | | $Y_x$ is the WY of grid $x$ (mm); $P_x$ is the annual average precipitation of grid $x$ (mm); $AET_x$ is the annual average actual evapotranspiration of grid $x$ (mm). | | | |
| | HQ (InVEST) | $Q_{xj} = H_J\left(1 - \left(\frac{D_{xj}^z}{D_{xj}^z + k^z}\right)\right)$ | (9) | 0.03 | |
| | | The total threat level for grid cell $x$ with land-use type $j$ is given by $D_{xj}$; $z$ ($z = 2.5$) and $k$ are scaling parameters (or constants); $H_j$ indicates the habitat suitability of land-use/cover type $j$. | | | |
| | CS (CASA) | $NPP(x,t) = APAR(x,t) \times \varepsilon(x,t)$ | (10) | 0.07 | [56] |
| | | $NPP(x,t)$ is the net primary production (gC·m$^{-2}$); $APAR(x,t)$ is the absorbed photosynthetically active radiation (gC·m$^{-2}$month$^{-2}$); $\varepsilon(x,t)$ is actual light energy utilization (gC·MJ$^{-1}$); $x$ and $t$ represent the spatial location and time, respectively. | | | |
| | WF (RWEQ) | $WE = \frac{2x}{S^2}Q_{max} \cdot e^{-(x/s)^2}$ | (11) | 0.31 | [57] |
| | | $Q_{max} = 109.8[WF \times EF \times SCF \times K \times (1 - C)]$ | (12) | | |
| | | $S = 150.71[WF \times EF \times SCF \times (1 - C)]^{-0.3711}$ | (13) | | |
| | | In the formula, $WE$ is the actual wind erosion amount (kg·m$^{-2}$); S is the regional $WF$ coefficient; $Q_{max}$ is the wind-sand retention (kg·m$^{-1}$); $C$ is the vegetation coverage; $Z$ is the calculated downwind distance (50 m); $WF$ is a meteorological factor; $EF$ is the soil erodibility factor; $SCF$ is the soil crust factor; $K'$ is the surface roughness factor. | | | |

The entropy approach is utilized in this article to summarize urbanization, LUCC, and ES indicators into different indicators [58,59]. The specific formula is as follows.

$$H(X) = -\sum_{i=1}^{n} p(x_i) \cdot log_2(p(x_i)) \tag{16}$$

$$W_i = 1 - \frac{H_i}{\sum_{i=1}^{n} H_i} \tag{17}$$

where $p(x_i)$ represents the probability of taking a value of $x$; $H(X)$ is the sum of all information entropy.

### 2.4.3. Coupling Coordination Degree

The coupling coordination degree is a popular tool for studying the coordination and coupling relationships between multiple systems [16]. In this study, the coupling coordination degree is used to determine whether the systems maintain a positive interaction and development. The arithmetic formula is as follows:

$$C = 3 \times \sqrt[3]{\frac{\varepsilon \times \omega \times \mu}{(\varepsilon + \omega + \mu)^3}} \tag{18}$$

$$T = \alpha\varepsilon + \beta\omega + \gamma\mu \tag{19}$$

$$D = \sqrt{C \times T} \tag{20}$$

where $C$ is the coupling coordination degree. $T$ is the comprehensive coordination index. $D$ is the degree of coupling. According to previous research classification standards, the coupling coordination degree is divided into five classes: 0.8–1 is high coordination; 0.6–0.8 is medium coordination; 0.4–0.6 is basic coordination; 0.2–0.4 is medium imbalance; and 0–0.2 is high imbalance [44,45].

### 2.4.4. Analysis of Driving Forces for Coupling Coordination Degree

(1)　Path analysis

Path analysis is a multivariate statistical method for investigating and validating the causal relationship between various system factors [46]. Compared with traditional correlation and regression analyses, it is able to more clearly reflect the potential feedback relationship between the elements.

(2)　Multiscale Geographically Weighted Regression (MGWR)

The multiscale reweighted regression method is an innovative method for analyzing spatial heterogeneity [47]. This methodology enables the optimization of variable-specific bandwidths, thereby improving the accuracy and precision of the results [48].

$$y = \sum_{j=1}^{k} f_i + \varepsilon \tag{21}$$

$$f_i = \beta_{bwj} x_{ij} \tag{22}$$

The $f_i$ is the residual. $\varepsilon$ is the initial residual calculated. $x_{ij}$ is the variant.

### 2.4.5. Future Projections of Coupling Coordination Degrees

(1)　Future scenarios setting of shared socioeconomic pathway/representative concentration pathway (SSP-RCP)

Various communities have developed SSP-RCP scenarios over the last decade [60,61]. This study employed four representative scenarios (Table 3) and incorporated the economic data that correspond to them [62].

(2)　PLUS Model

The PLUS model employs cellular automata (CA) and a random forest classification algorithm. Compared with the traditional CA-Markov method, it is able to predict patch changes in land more accurately [49], especially for forest and urban patches [50]. The PLUS model's simulation process includes land-use expansion extraction, land-use expansion

analysis using the Land Use Expansion Analysis Strategy (LEAS), and a cellular automata model based on multi-class random patch seeds (CARS). The PLUS model was used in this study to forecast future land use. This study incorporated future land-use data, which were derived from the Global Change Assessment Model (GCAM). The integration was achieved using spatial statistics to determine land-use requirements. The Kappa coefficient of 0.8463 showed that the simulations were highly accurate.

(3)　　Random forest model

Random forest is an integrated approach. It uses trees as base learners and combines their predictions by averaging. Random forest has good utility performance and has more stability than traditional regression methods [63,64]. In this study, random forest regression was utilized to predict the ES index for the year 2035 by integrating future land-use simulations with future socioeconomic and other relevant data. The number of decision trees used in this algorithm was determined to be 500 after the parameter test. R 4.2.2 was used to implement the algorithm. This model's training accuracy was 0.986, while its verification accuracy was 0.894. The random forest model's primary computation formula is as follows [51]:

$$\bar{h}(x) = \frac{1}{T} \sum_{t=1}^{T} \{h(x, \theta_t)\} \tag{23}$$

where $h(x, \theta_t)$ is the output of variable $x$; $T$ is the number of base learners.

**Table 3.** Scenario descriptions.

| Scenario | Description |
| --- | --- |
| SSP1-2.6 | Combination of low societal vulnerability and low forcing level, with significant land-use change |
| SSP2-4.5 | Combination of intermediate levels of societal vulnerability and intermediate levels of forcing |
| SSP4-6.0 | Combination of relatively high societal vulnerability and relatively high forcing level, with significant land-use change |
| SSP5-8.5 | Combination of high societal vulnerability and high level of forcing |

## 3. Results

### 3.1. Spatiotemporal Variation in Urbanization, LUCC, and ESs in CPAs

This study used cluster analysis to divide 14 CPAs into three sub-regions based on indicators of urbanization, LUCC, and ESs. The classification results are shown in Figure 3.

#### 3.1.1. Spatiotemporal Variation in Urbanization

During the research period, CPAs' urbanization increased significantly, and it continued to increase after 2013 (Figure 4). Between 2000 and 2018, the level of spatial urbanization remained relatively stable. The population's urbanization increased gradually and then declined. According to statistical data, while the proportion of the urban population was gradually increasing, the total population decreased significantly. The economic urbanization and urbanization indexes significantly improved, particularly around 2013. In terms of regional urbanization differences, there were no discernible differences in the urbanization levels of the three regions' populations, and there were no significant trends in space (Figure S2). The SN area had significantly lower levels of spatial urbanization. In terms of population urbanization, all regions saw a significant decrease in 2018, with the SN seeing the greatest decrease.

#### 3.1.2. Spatiotemporal Variation in Land

Table 4 shows the changes in various land types between 2000 and 2018. The forest area grew steadily, with an average annual growth rate of 0.14%. However, cropland, grassland, and unused land all decreased to varying degrees during this time period. Construction

had the highest growth rate, with a K value of 3.28%. The intensity of LUCC was higher in the NN and SN but lower in the WN (Figure 5). In all regions, LUCC primarily involved the conversion of cropland, forest, grassland, and water. The conversion from grassland to water was most noticeable in the NN, with the greatest shift in the WN occurring from unused land to grassland and the greatest shift in the SN occurring from unused land to water (refer to Table S1).

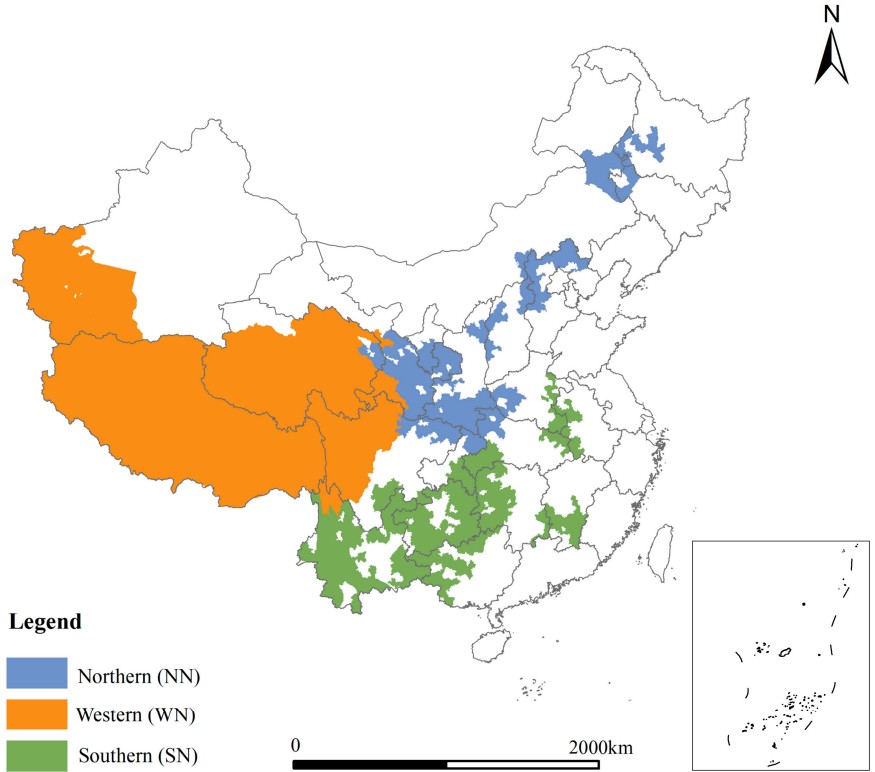

**Figure 3.** CPA partition results. NN includes DXAL, LL, LP, QB, and YS regions; WN includes SSZQ, XZ, and XJ regions; SN includes DG, DX, WL, and WM regions.

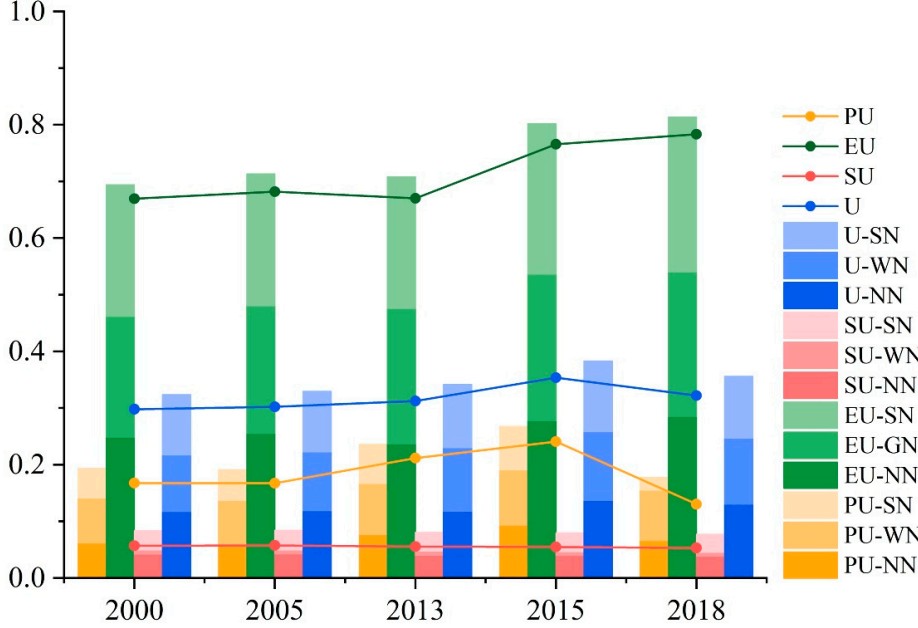

**Figure 4.** Indicators of urbanization over time. PU stands for population urbanization; SU stands for spatial urbanization; EU stands for economic urbanization; and U stands for urbanization.

**Table 4.** The K value and area of each land-use type.

|  | K | 2000 | 2005 | 2013 | 2015 | 2018 |
|---|---|---|---|---|---|---|
| Cropland | −0.22% | 427,675 | 417,734 | 413,331 | 411,574 | 408,992 |
| Forest | 0.14% | 969,300 | 974,465 | 985,181 | 986,276 | 991,597 |
| Grassland | −0.04% | 1,799,458 | 1,799,457 | 1,788,104 | 1,793,255 | 1,792,966 |
| Water | 1.16% | 51,370 | 55,984 | 60,577 | 60,390 | 62,092 |
| Unused land | −0.12% | 66,490 | 66,316 | 65,984 | 65,474 | 64,922 |
| Construction | 3.28% | 12,397 | 13,684 | 17,445 | 18,246 | 19,621 |

The unit is km$^2$.

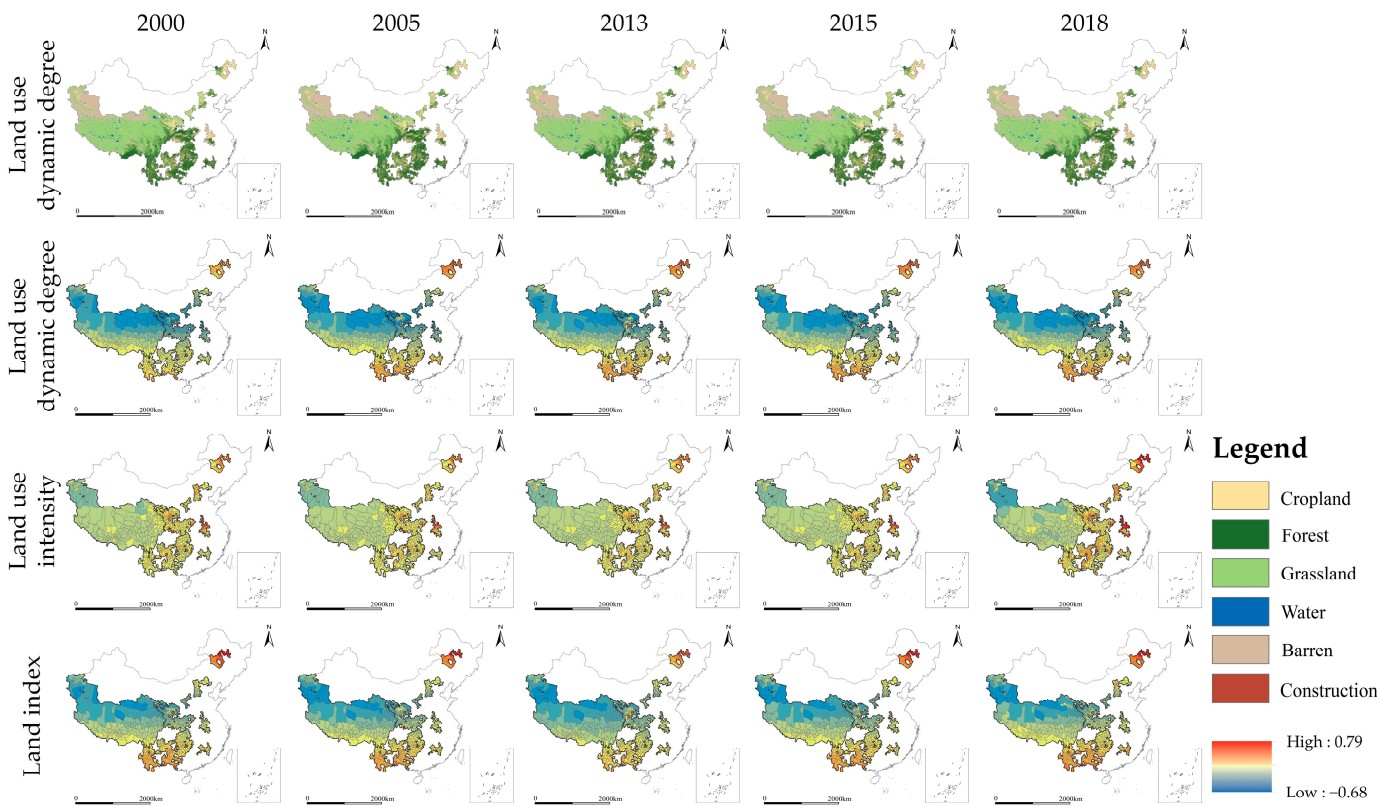

**Figure 5.** Spatial distribution of land indicators.

In terms of land-use intensity, human interference was more severe in the eastern and northeast regions. The overall spatial trend is positive in the east and negative in the west. The WN area and the other two areas had very distinct spatial differences, while the NN was comparable to the SN. Overall, LUCC and human interference were relatively strong in the eastern and southern regions. LUCC and human interference were low in the northern part of the WN, indicating greater development potential.

### 3.1.3. Spatiotemporal Variation in ESs

The ES index exhibited a spatial pattern characterized by lower values in the northern regions and higher values in the southern regions. Around 2013, the trend shifted from declining to rising, but it was still on the rise overall. From northwest to southeast, CS and HQ showed gradually increasing trends (see Figure 6). These patterns remained consistent over time. WY increased from north to south, especially in the Himalayas. SC remained constant. WF's high-value areas gradually expanded over time. With the exception of the SN, the trend of changes in the ES index was consistent across all regions. The only area in which it continued to rise was the SN. CS in the SN consistently increased, particularly between 2005 and 2013. (See Table S2). The SN had the highest levels of CS, SC, and WY throughout the study period. In contrast, the HQ in this region had a consistent negative

trend. WF showed fluctuating but gradually increasing trends, according to the findings, with the exception of a more obvious decline in the NN. The observed change patterns suggest a trend in which high-value areas decreased while low-value areas increased, resulting in a reduction in regional disparities. In summary, there was a noticeable change in the pattern of ES trends around 2013.

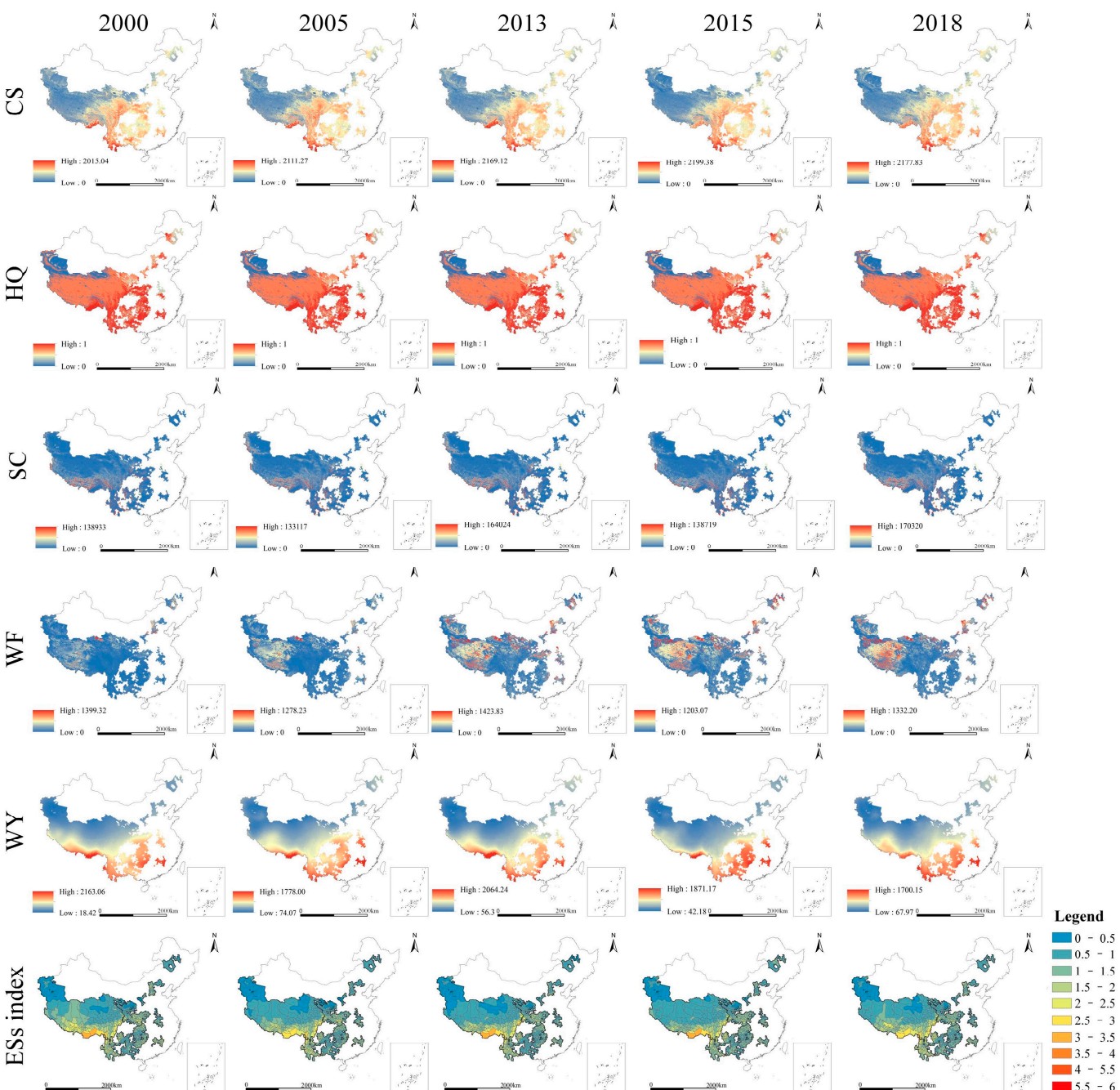

**Figure 6.** ES spatial distribution in CPAs in 2000, 2005, 2013, 2015, and 2018. Blue has a low value, whereas red has a high value. The unit of CS is kgC·m$^{-2}$; HQ is dimensionless and ranges from 0 to 1; SC and WF units are t·hm$^{-2}$; the unit of WY is mm.

### 3.2. Coupling Coordination Relationships among Urbanization, LUCC, and ESs

From Figure 7, the coupling coordination degree of the sub-regions fluctuated in a downward trend. The shift in coupling coordination corresponded more closely to the shift in the comprehensive coordination index. The NN's and WN's coupling coordination degrees were reduced from basic coordination to medium imbalance, with the NN

decreasing from 0.42 to 0.38 and the WN decreasing from 0.40 to 0.35. In addition, their coupling degree and comprehensive coordination index decreased. The aforementioned trend was most visible between 2005 and 2013, with changes occurring between 2013 and 2015. However, the SN always remained at the basic coordination level (0.42). According to the level of coupling coordination in the county, most areas were at the basic level. The area of medium imbalance significantly decreased, while the area of medium coordination significantly increased. Overall, the level of coordination tended to improve gradually.

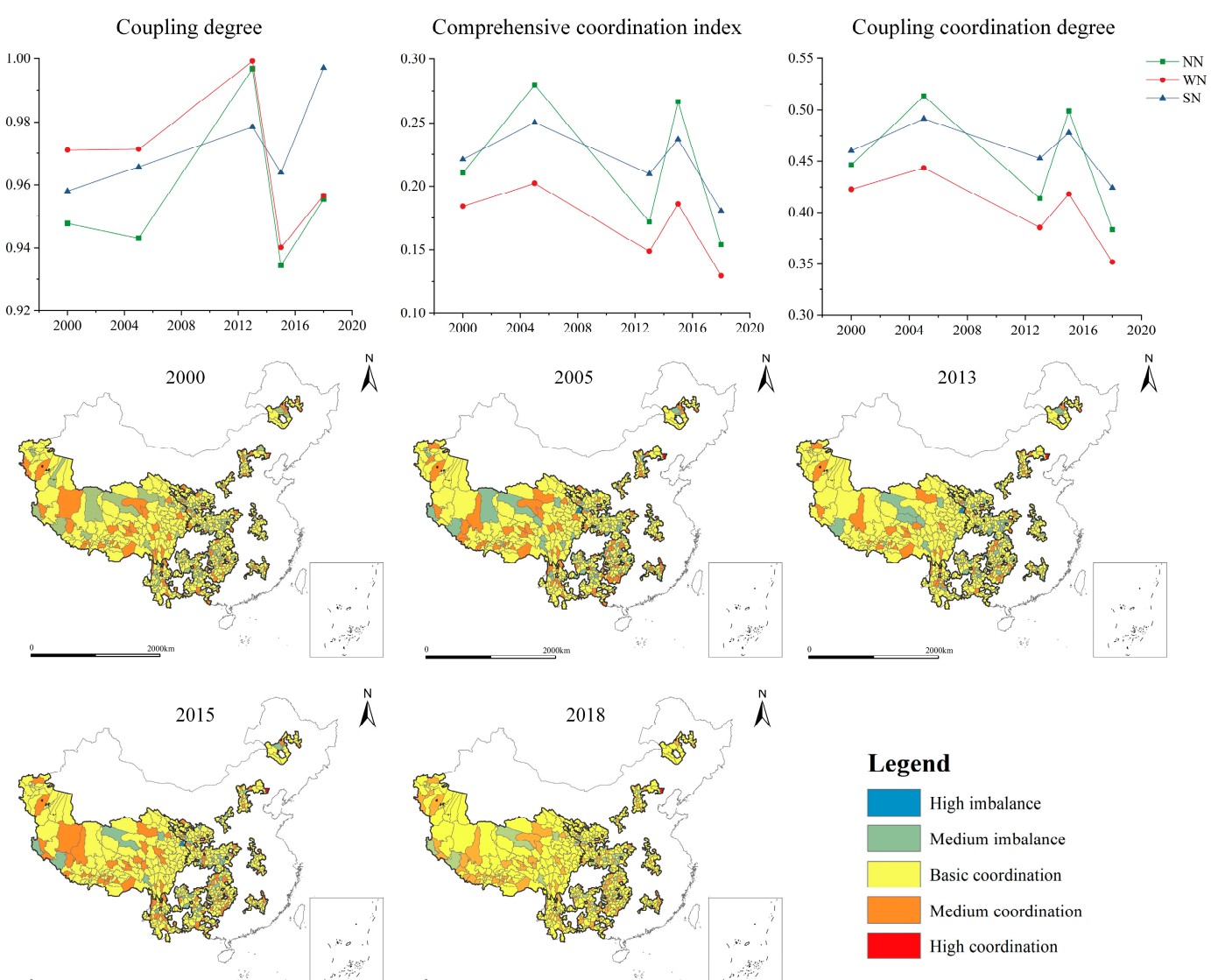

**Figure 7.** Spatial distribution of coupling coordination degree.

According to Figure 8, it is clear that the three systems had a complex feedback and negative feedback relationship. The effects of urbanization on LUCC and ESs were profound. The impact of urbanization on LUCC was unidirectional, and the change in ESs could be driven by the impact on LUCC. The land-use type was found to be closely related to ESs, with a standardization coefficient of 0.841. Secondly, ESs had a negative feedback relationship with urbanization and land, and the size of the coefficient was comparable to the forward coefficient of action. The various indicators of urbanization, grassland, unused land, and geographical factors all had an impact on the system's coupling and coordination. The influence of the coupling coordination of urbanization indicators was positive among them. In other words, urbanization–LUCC–ESs were in a state of mutual constraint equilibrium. The extension of grassland and a decrease in unused land benefited



the coupling coordination degree, particularly in the WN. Geographically, the districts and counties that had an adverse influence on the coupling coordination degree were concentrated in the northwest. Positive effects predominated in the eastern part of the WN and other regions.

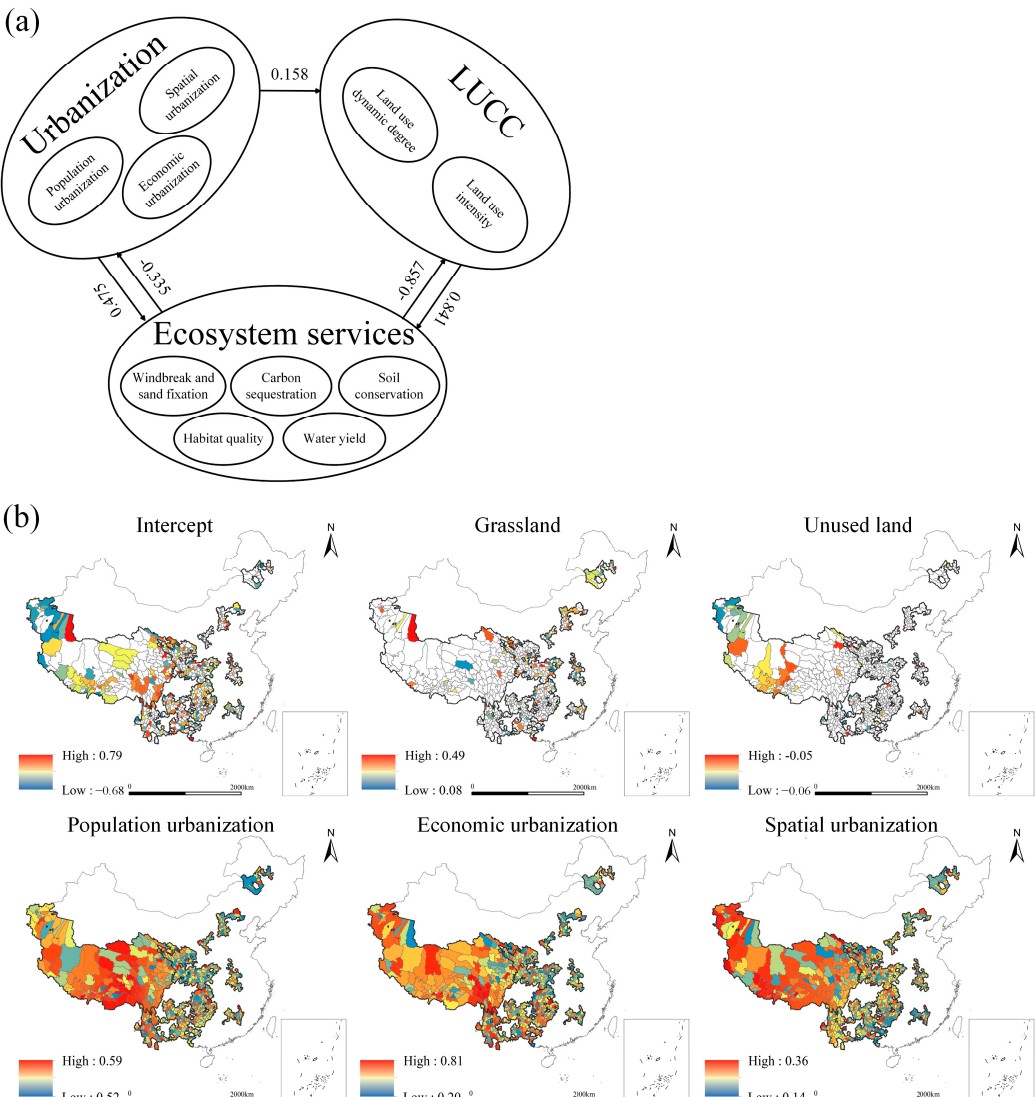

**Figure 8.** (**a**,**b**) represent the relationship between urbanization, LUCC, and ESs and the driving force analysis of the coupling coordination degree based on the MGWR model.

*3.3. Prediction under SSP-RCP Scenarios*

Under each scenario, the area of grassland and construction is expected to increase significantly by 2035. These increases are primarily due to a decrease in cropland, unused land, and forests (Figure 9). Grassland areas with medium ecological value have promising future development prospects. Cropland conversion has exacerbated forest erosion, resulting in the loss of high-value ecological areas. Notably, the forest and grassland areas in SSP1-2.6 differ significantly from those in the other scenarios, whereas the differences between the remaining scenarios are less pronounced. Table S4 and Figure S3 show the spatial distribution of land use.

Except for the WN, other regions showed an upward trend in all future SSP-RCP scenarios in terms of coupling coordination changes (see Figure 10). In SSP1-2.6, the WN area only increased by 0.43. Each region demonstrated the basic coordination level in the context of SSP1-2.6. In other cases, the WN area is at a medium imbalance level and performs poorly. According to the changes in the county area's coupling coordination

degree, there are no county areas with a high imbalance in the SSP1-2.6 situation. From SSP1-2.6 to SSP5-8.5, the medium coordination area gradually decreased. Overall, SSP1-2.6 is the most plausible future development model for the CPAs. The SN varies less across scenarios, particularly in the high-emissions scenario, whereas the WN requires more attention and assistance to improve its coupling coordination level.

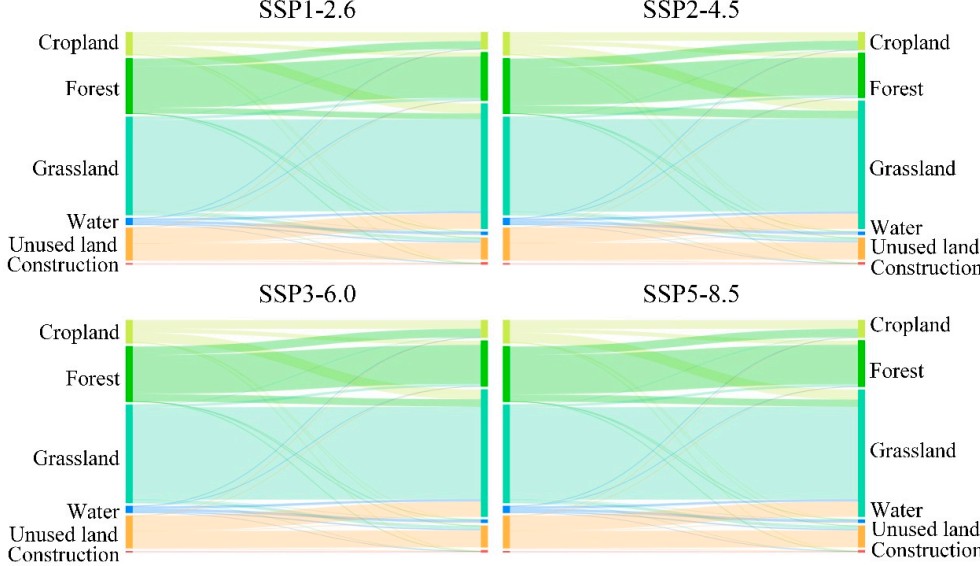

**Figure 9.** Future LUCC under different scenarios.

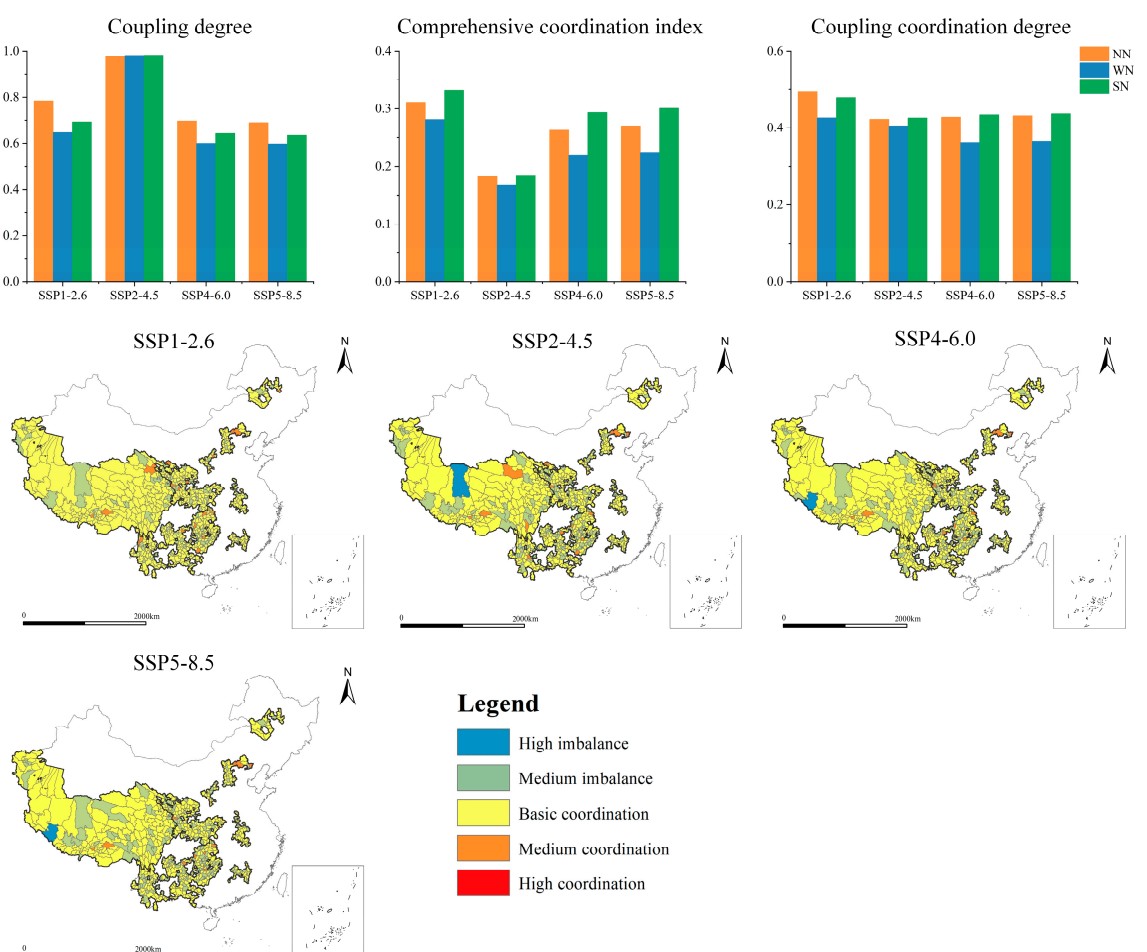

**Figure 10.** Distribution of coupling coordination degree in 2035 under different SSP-RCP scenarios.

## 4. Discussion

### 4.1. Key Role of Policy Regulation in Mitigating the Impact of Urbanization on ESs in CPAs

The general consensus has been that urbanization and policy-driven economic development led to land-use and land-cover changes, resulting in ecological impacts [11,12]. According to Table 5, the economic urbanization development pattern does not currently result in the case of sprawling construction or environmental deterioration for CPAs. Preceding the poverty alleviation policy, ESs experienced a downward trend, aligning with the pre-period characteristics of the EKC theory. However, after policy support was implemented, ESs displayed an upward trend, consistent with the later EKC theory characteristics. Policy support expedites the growth of secondary and tertiary industries, hastening industrial upgrading and attenuating environmental sacrifices. This acceleration reduces the time required to enter the platform period [65–67]. The annual rate of change for ecological land before and after 2013 showed an increasing trend for forests and water. Although the amount of construction is increasing, it still represents a small proportion of the total land. The increase was caused primarily by the encroachment of neighboring cropland (Table S1). When comparing ES changes before and after 2013, it is clear that SC, WF, and WY showed decreasing trends in most areas prior to 2013. The exception was HQ, which remained stable in most areas after BRI, while SC, WF, and WY showed the opposite trend (Table S2).

**Table 5.** Comparing the impacts of policy adjustments on urbanization, LUCCs, and ESs around 2013.

|  | **Before 2013** | **After 2013** | **Explanation** |
|---|---|---|---|
| Urbanization | 0.0004 | 0.0007 | The rate of urbanization has accelerated significantly. While population urbanization has slowed slightly, economic urbanization has accelerated. The implementation of targeted poverty alleviation policies, as well as the industrial restructuring strategy, has produced visible results [68]. The role of spatial urbanization, on the other hand, is not obvious. |
| LUCC | 0.0056 | 0.0088 | The K value for forest has almost tripled since 2013, whereas the K value for construction has doubled. Despite growth being observed on both ecological and developed lands, the rate and magnitude of expansion in ecological areas significantly outpaces those in developed areas. The proportion of construction remains relatively insignificant. |
| ESs | −0.0015 | 0.0023 | The majority of the increase in ESs is attributable to the rise in SC, WF, and WY. The general trend of ESs is a shift from decreasing to rising. In addition to the role of climate change, the expansion of ecological land use (particularly forests) has had positive effects on a global scale [5,69]. However, the ecological value of additional land is limited, and early forest conservation policies impeded the development of ecosystem services such as CS [70]. |

Note: The numbers in Table 5 represent the annual average change in the respective indexes.

### 4.2. Determinants Influencing Coupling Coordination within CPAs

Although the degree of coupling coordination has not decreased significantly, it remains low (Figure 9). Some previous similar studies agree with this [71,72]. In accordance with the traditional view of economic underdevelopment [73], the primary cause of this issue can be attributed to the slow pace of urbanization and low intensity of land use. Although urbanization in CPAs is gradually increasing, it focuses primarily on economic aspects and has minimal impacts on ecological land. Consequently, urbanization and LUCC continue to have positive effects on ESs. It should be noted, however, that ESs exert a suppressive influence on urbanization and LUCC (Figure 8). The effects of urbanization development, LUCC, and ESs are subject to various urbanization and LUCC impact patterns. To improve sustainable development, a stable and enabling land-use environment is essential. This presents constraints and obstacles to both space urbanization and LUCC. Consequently, the connection among urbanization, LUCC, and ESs emphasizes the central role of land as a linking system.

In terms of the factors that influence coupling coordination, construction expansion has a negligible effect, and grassland is a critical factor influencing the coordinated growth of various sectors. Whether viewed from the standpoint of its importance to the ecosystem or of its coordinated development, grassland is the type that requires special attention [74]. As a low-value type of land, it is critical to keep unused land from eroding grassland, particularly in areas that are geographically like WN. Therefore, in the future process of rural revitalization and new urbanization, the coordinated maintenance of grassland resources and the rational allocation of construction land and ecological land are essential for sustainable development.

### 4.3. Policy Recommendations

Due to the complex geography and continuing economic challenges, it is necessary to create new urbanization plans for CPAs that are adapted to the local situation. Although ESs are developed in conjunction with urbanization, the disparity is still gradually increasing. According to projections, the forest ecosystem will face significant threats. It is necessary to adhere to the red lines of ecological protection, permanent basic cropland, and urban development boundaries as strict spatial constraints for the protection and promotion of sustainable development [75,76]. Furthermore, CPAs represent a specialized area of management, necessitating a departure from traditional provincial and regional approaches in favor of differentiated land-use management guided by zoning classification principles and comprehensive coordination. (1) The NN. Human activities in the NN area have had a significant impact on LUCC, leading to increased conflicts over land resources. As a result, it is critical to prioritize ecological restoration efforts when considering coordinated development in the NN, especially in forests and grasslands. The improvement of land efficiency and the implementation of intensive management should be further emphasized. (2) The SN. The SN area has the most balanced urbanization–land–ES relationship of the three areas. However, the topography and geomorphology of the SN region are relatively complex, and there are large areas of ecological fragility [77]. The importance of preservation over restoration is critical for preserving the ecological integrity [78]. It is suggested that the area be designated as a pilot region for policies to harmonize ecological and economic development while establishing a comprehensive green economic structure encompassing sectors such as eco-tourism and an environmental protection economy. (3) The WN. The economic and ecological value of land in the WN area is relatively low. As a result, effective ecological protection mechanisms are critical to achieving coordinated development. Special attention should be paid to protecting the transitional zone between grassland and unused land, as well as implementing a more sensible grazing management program.

To circumvent the early-stage environmental sacrifices often associated with the EKC theory and progress toward synergistic development phases, a strategic approach to land management that aligns with local conditions is imperative. Firstly, it is vital to encourage other developing nations to adopt sound land-use practices, upgrade their industries, and foster environmentally friendly economic models. Secondly, implementing pilot policies in environmentally favorable regions while prioritizing environmental protection in less favorable areas is crucial. Emphasis should be placed on LUCC in marginal land zones across various land types and in ecologically fragile areas to prevent further encroachment upon valuable ecological territories. Lastly, leveraging international cooperation and policy support becomes pivotal in accelerating the establishment of special economy-supportive zones in economically disadvantaged regions while mitigating the environmental repercussions of conventional developmental processes.

### 4.4. Innovativeness and Limitations

In comparison to previous studies, this study offers an assessment and comparison of the changes in coupling coordination degrees in CPAs at both the county scale and by region. The PLUS model is more suitable for the prediction of patchy land, which is consistent with the spatial distribution of land use, whereas the random forest model

is more accurate and stable. Potential large errors in the data can be eliminated. The integration of the PLUS model with the random forest model facilitates the prediction of future coupling coordination degrees, addressing a research gap in poverty areas at the county level. Previous studies have predominantly focused on specific indicators like land [79] or carbon sequestration [80], with limited exploration of coupling coordination prediction in an integrated manner. By employing SSP-RCP scenarios, the combined use of the PLUS model and random forests synthesizes previous research, providing a novel policy-planning approach for economically disadvantaged regions facing dual challenges of economic development and environmental protection. Furthermore, LUCC appears more pronounced in non-poor areas than in impoverished regions, and the statistics regarding urbanization and environmental pollution are more comprehensive. Therefore, urbanization and social indicators in non-poverty areas should be expanded upon. Particularly in highly urbanized regions, considerations such as urban pollution and resident well-being should be factored in for more targeted and coordinated developmental analyses.

However, there are certain limitations in this study. Firstly, the majority of the examined ESs in this study primarily focus on regulation services, with limited exploration of others. Secondly, the utilization of low-resolution meteorological data and uncertainties in the InVEST model's calculation methods hinder a comprehensive consideration of soil, temporal changes, and entire ecosystems [81,82]. Although Figure S1 indicates adequacy for large-scale monitoring, disparities in the investigated values persist. Moreover, China's current demographic and urbanization transitions introduce significant uncertainties in projection data based on SSP-RCP scenarios, a global challenge under active research. Future studies should encompass a broader spectrum of ecosystem services, construct more precise assessment models, and enhance the accuracy of predicted foundational data.

**5. Conclusions**

This work evaluates the interrelationships among urbanization, LUCC, and ESs in CPAs and their spatiotemporal variation characteristics and projects the coupling coordination degree under different SSP-RCP scenarios in 2035 using the coupling coordination degree model, path analysis model, MGWR model, PLUS model, random forest model, and SSP-RCP scenario projection. The study provides a comprehensive perspective for understanding the degree of coupling coordination at the county scale in CPAs. The study can be concluded as follows:

(1) Urbanization, LUCC, and ESs in China's CPAs have shown inconsistent upward trends. Before 2013, ESs had a slight decline overall, which is consistent with the early phase of the Environmental Kuznets Curve (EKC). After 2013, it was characterized by a later phase. The EKC theory is only valid in the short term. Urbanization and LUCC growth accelerated after 2013, especially in the NN region. ESs were most pronounced in the SN.

(2) Over time, the coupling coordination degree in CPAs slightly diminished. LUCC was essential for maintaining system balance. The SN remained relatively stable and at the basic coordination level. The western and northern regions reached a medium imbalance. The degree of coupling coordination was primarily influenced by urbanization, geographical factors, grassland, and undeveloped land.

(3) Under each SSP-RCP scenario, the coupling coordination degree in CPAs exhibits an upward trend in 2035. According to the SSP1-2.6 scenarios, environmental prioritization and sustainable routes are the best approaches for future CPA development.

(4) Environmentally friendly urbanization demands thoughtful land management tailored to local conditions, emphasizing long-term environmental protection. The SN region is highly coordinated and should establish an ecological pilot zone; the WN region should prioritize the protection of the original grassland ecosystem; and the NN region should improve the efficiency of land management, promote land intensification, and focus on the implementation of ecological restoration.

In summary, contrary to prevailing beliefs, CPAs are undergoing rapid urbanization without significant ecological damage. However, it should be noted that the applicability of the EKC theory in CPAs is limited in the short term due to poverty and ecological policies. The coupling coordination degree will increase further in 2035. However, the assessment and prediction of coupling coordination degrees reveal potential inconsistencies between urbanization and ecological conservation. Therefore, it is imperative to develop reasonable land-use planning and policy support based on local natural and socioeconomic characteristics. This study provides sub-regional policy recommendations to inform sustainable development in developing countries and poor regions. Future research will further optimize monitoring techniques and research methods to enrich the knowledge of ecosystem services.

**Supplementary Materials:** The following supporting information can be downloaded at https://www.mdpi.com/article/10.3390/land13010082/s1, Table S1. Sub-regional land use transfer matrix. Table S2. The average change in ESs. Table S3. MGWR model index. Table S4. Future land prediction under different SSP-RCP scenarios. Figure S1. a–e are linear regression scatter plots of carbon sequestration, habitat quality, soil conservation, windbreak sand fixation, and water yield and verification data. Figure S2. Spatial distribution of urbanization indicators. Figure S3. Land use distribution in 2035 under different SSP-RCP scenarios.

**Author Contributions:** Conceptualization, J.Z. and D.Y.; Methodology, J.Z. and X.L.; Software, X.L.; Formal analysis, X.L.; Writing–original draft, X.L.; Writing–review & editing, J.Z., Y.Q., Y.Z. and D.Y. All authors have read and agreed to the published version of the manuscript.

**Funding:** This research received no external funding.

**Data Availability Statement:** Data is contained within the article.

**Conflicts of Interest:** The authors declare no conflict of interest.

## Abbreviations

| | |
|---|---|
| CPAs | Contiguous poverty areas |
| LUCC | Land-use and land-cover change |
| ESs | Ecosystem services |
| MGWR | Multiscale geographically weighted regression |
| PLUS | Patch-generating Land Use Simulation Model |
| SSP-RCP | Shared socioeconomic pathway/representative concentration pathway |
| NN | Northern contiguous poverty areas |
| WN | Western contiguous poverty areas |
| SN | Southern contiguous poverty areas |
| CS | Carbon sequestration |
| HQ | Habitat quality |
| SC | Soil conservation |
| WF | Windbreak and sand fixation |
| WY | Water yield |

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
