# Peer review of "Can Urbanization-Driven Land-Use and Land-Cover Change Reduce Ecosystem Services? A Case of Coupling Coordination Relationship for Contiguous Poverty Areas in China"

_land, doi:10.3390/land13010082_

Round 1

Reviewer 1 Report

Comments and Suggestions for Authors

This is an article with a traditional approach, but with some practical implications. Overall, the essay is well-structured and has no major problems other than being slightly less innovative. Therefore, I recommend minor repairs. The suggestions are as follows, hoping to further improve the quality of the article.

1.     The second paragraph of the introduction needs to be further condensed, rather than listing who researched what.

2.     Line 117-118. CPAs had a population of 3.13 million people in 2019, with a per capita disposable income of about $1,634, which is less than 20% of the national average of about $10,103.What is the data source?

3.     How were the weights in Table 2 determined?

4.     China is currently in a transitional period of population change and urbanization. Therefore, there may be significant uncertainty in the 2035-related data provided in the database. Suggest the author to further explain in the discussion.

Comments on the Quality of English Language

Minor editing of English language required.

Reviewer 2 Report

Comments and Suggestions for Authors

In this paper, the authors evaluated the interrelationships among urbanization, land use and land cover change, and ecosystem services in contiguous poverty areas and the spatial and temporal variation characteristics, and projected the coupled coordination degree under different SSP-RCP scenarios in 2035, by the coupled coordination degree model, path analysis model, MGWR model, PLUS model, random forest model, and SSP-RCP scenarios projection. This paper provided recommendations and support for regional land sustainable development management decisions in the context of urbanization. The findings are also very interesting and clearly presented. However, I have some reservations. 1. The abstract needs to refine the research conclusions according to the research results. The SSP1-2.6 scenario is the best option for CPAs’ development? The innovation of this paper is not clear enough. 2.The necessary discussion with the previous research results is lacking. Most previous research has concentrated on country and city-level studies, and understanding of the complex relationship between urbanization and LUCC-ESs remains oversimplified. Some scholars have already paid attention to this topic that the spatiotemporal variation of urbanization, LUCC, and ESs, but the shortcomings of previous research, as well as the differences between this study and theirs. 3. Figure 5. Spatial distribution of land indicators definitely requires further development. Please supply the basin basis method. 3.1.2. Spatiotemporal Variation of Land or land use types LUCC? The figure is not self-evident enough and the font format is not well. 4. In the discussion part, the meteorological data used in the article are of low resolution, and there are uncertainties in the calculation methods of the In VEST model, which should be connected with the relationship with the previous model analysis. The progress of future research will be more convincing. 5. The research conclusion section needs to be re summarized and refined. Through the research in this paper, what scientific issues have been discovered, and what new research results and conclusions have been obtained

Comments on the Quality of English Language

English expression can be appropriately enhanced or polished.

Reviewer 3 Report

Comments and Suggestions for Authors

Under the context of ecosystem services function decline, it is very necessary to study the effect of urbanization and LUCC. This paper evaluates the coupled and coordinated relationship between urbanization and LUCC-ESs. It provides a reference for understanding the spatiotemporal variation of urbanization and promoting the sustainable development of contiguous poverty areas. I think this is a good manuscript that uses appropriated methodology and rigorous analysis.

 -I suggest a better description of the limitation or research gap of current studies in the section of “Introduction”. More findings and discussions from the literature review should be presented.

 -References should be avoided in the Discussion section.

 -The conclusions are broad, but they bring together the main results of the research. However, the question remains if the methodology is replicable for other Non-poverty areas or if some adjustments have to be made? And how could the results be validated? Perhaps the authors will have the opportunity to read the paper again and try to emphasize these points in the methodology or discussion.

Comments on the Quality of English Language

I have no comments on the quality of English language.

Reviewer 4 Report

Comments and Suggestions for Authors

The manuscript is well written and the topic is really interesting.

However, major revisions are needed:

1. Importance and necessity. Please clearly explain the importance and necessity of the research in the paper in Introduction Section.

2. Scientific originality/novelty: The novelty/originality should be clearly justified that the manuscript contains sufficient contributions to the new body of knowledge from the international perspective. What new things (new theories, new methods, or new policies) can the paper contribute to the existing international literature? This point must be more reasonably justified by a Literature Review, clearly introduced in Introduction Section, and completely discussed in Discussion Section.

3. Theoretical contribution: The theoretical thinking in the paper should be strengthened. Which theoretical or academic arguments are the authors aiming to examine or discus? What new theories or theoretical knowledge this paper can add in the existing international literature by looking the case study? All these questions must be clearly responded in Introduction, Analysis and Discussion Section.

4. Please, explain more about the methods in section 2.2. Research Framework, adding specific references.

5. Global knowledge contribution: This paper fails to engage with the wider readership of the journal. What new global knowledge can this paper contribute to the existing international literature? How to link the findings and conclusions in this paper with the previous findings and conclusions from other countries?

6. A thorough and criticism-featured Literature Review section is needed. The existing literature review is insufficient.

7. It would be useful to discuss in more depth the differences and similarities of your method with other methods that have the same purpose and explain why you consider it more suitable to adopt. The case study is certainly useful to apply the method and better explain the possible results, but you need to go beyond the application, explaining why and how it could be used in other realities.

8. Clear urban policy or planning concerns should be expanded in this paper. Firstly, policy issues should be clearly introduced in introduction section. Secondly, policy arguments or debates should be further discussed in analysis and discussion section. Thirdly, Takeaway for Practice is also encouraged to be included in this paper. It should be clear enough to present your policy recommendations for both local and international practice.

Round 2

Reviewer 4 Report

Comments and Suggestions for Authors

The changes made (both in response to my requests and those of the other reviewers) are sufficient to make the paper publishable. 
